# Low-level buoyancy as a tool to understand boundary layer transitions

Francesca M. Lappin[1,2], Tyler M. Bell[2,4], Elizabeth A. Pillar-Little[1,2], and Phillip B. Chilson[1,3]

[1]School of Meteorology, University of Oklahoma, Norman, OK, USA
[2]Cooperative Institute for Severe and High-Impact Weather Research and Operations, Norman, OK, USA
[3]University of Oklahoma Advanced Radar Research Center, Norman, OK, USA
[4]NOAA/OAR National Severe Storms Laboratory, Norman, OK, USA

**Correspondence:** Francesca Lappin (francesca.lappin@ou.edu)

**Abstract.** Advancements in remotely piloted aircraft systems (RPAS) introduced a new way to observe the atmospheric boundary layer (ABL). Adequate sampling of the lower atmosphere is key to improving numerical weather models and understanding fine-scale processes. The ABL's sensitivity to changes in surface fluxes leads to rapid changes in thermodynamic variables. This study proposes using low-level buoyancy to characterize ABL transitions. Previously, buoyancy has been used as a bulk parameter to quantify stability. Higher resolution data from RPAS highlight buoyancy fluctuations. RPAS profiles from two field campaigns are used to assess the evolution of buoyancy under convective and stable boundary layers. Data from these campaigns included challenging events to forecast accurately, such as convection initiation and a low-level jet. Throughout the daily ABL transition, results show that the ABL height determined by the minimum in vertical buoyancy gradient agrees well with proven ABL height metrics, such as potential temperature gradient maxima. Moreover, in the cases presented, low-level buoyancy rapidly increases prior to convective initiation and rapidly decreases prior to the onset of a low-level jet. Low-level buoyancy is a force sensitive in space and time, and with further analysis could be used as a forecasting tool. This study expounds on the utility of buoyancy in the ABL and offers potential uses for future research.

## 1 Introduction

The atmospheric boundary layer (ABL) is strongly influenced by kinematic and thermodynamic interactions with the Earth's surface. It is sensitive to changes in radiation, low-level moisture, and heat fluxes. The ABL functions as a conduit for moisture and momentum to be transported vertically. As a consequence, the depth of the ABL (referred to herein as ABL height) and the ABL stability fluctuate in time and space (Lenschow et al., 1979; Stull, 1988). This influences local weather (Lapworth, 2006), turbulence (Banta et al., 2003; Bonin et al., 2013), and aerosol transport (Nilsson et al., 2001; De Wekker et al., 2009; Pal et al., 2014). The nature of the ABL makes it both crucial to successful numerical weather prediction (NWP) yet incredibly difficult to represent. Most boundary layer parameterizations are based on observation methods that are decades-old. Often, the best choice for boundary layer parameterizations is situationally dependent to what is being modeled (Braun and Tao, 2000; Nolan et al., 2009; Hu et al., 2010; Cuchiara et al., 2014; Cohen et al., 2015). Weather and climate models will continue to struggle to accurately represent the ABL without vertical, high-resolution observations (Steeneveld et al., 2008; Teixeira et al., 2008;

Baklanov et al., 2011). Assimilating in situ data has been shown to benefit the performance of the model (Ruggiero et al., 1996;

Otkin et al., 2011; Jonassen et al., 2012; Ágústsson et al., 2014; Reen et al., 2014; Jones et al., 2016; Degelia et al., 2018). The lack of accessible technology to accurately sample the ABL has slowed advancements throughout the field. Up until recently, these types of data have not been easily retrievable.

In the past, it has proven difficult to collect adequate spatially and temporally resolved measurements within the ABL, resulting in a "data gap". Since the National Research Council (2009) called for more vertical measurements in the ABL, there have

been technological advancements to address the gap. Remote sensors such as microwave radiometers, lidars, and scatterometers can continuously measure the lower atmosphere, which have been shown to improve short-term forecasts (Coniglio et al., 2019; Hu et al., 2019; Lewis et al., 2020). However, these instruments are expensive and typically need to be used in tandem to get a complete sample. Most remote sensors are mobile but not nimble, which limits the environments they can sample. For example, pre-convection environments change rapidly, and instruments need to relocate quickly to gain targeted observa-

tions. Another tool more commonly used to capture ABL measurements is the radiosonde which are typically released twice daily across the United States. Their upper-air measurement aid greatly in seeing synoptic patterns throughout the troposphere. Unfortunately, radiosondes are only released frequently enough to capture mesoscale changes during field experiments. The spatial and temporal frequency of radiosonde release is inadequate for convection allowing models. Of equal importance, their spatial resolution through the ABL is too coarse for thorough characterization. While there are avenues to shrink the data gap,

we still lack an infrastructure to address this on a broader scale.

Increasing interest in remotely piloted aircraft systems (RPAS) across many sectors has accelerated improvements in quality and availability of RPAS technology (Reuder et al., 2009; Elston et al., 2015; Villa et al., 2016). In turn, the capabilities of RPAS broadened and its usefulness in research became obvious. The benefits of utilizing RPAS in atmospheric sciences have been proven across many situations, including turbulence observations and data assimilation (Dias et al., 2012; Båserud et al.,

2016; Flagg et al., 2018; Barbieri et al., 2019). RPASs can be readily reused to sample rapidly changing environments, and can be paired with remote sensors to describe the lower troposphere more completely. In Bell et al. (2020) measurements from RPAS, radiosondes, and remote sensors were found to agree well with each other. The study also discusses functionality of each platform. While RPASs allow for more adaptive sampling, there are more federal regulations and air restrictions governing their use. Nonetheless, some specially designed RPASs can deliver equally accurate measurements as radiosondes with a higher

spatial resolution and are more cost-efficient. The confidence shown in the data collection and usefulness in data assimilation will prove its place as a reliable observation platform. RPASs stand as an affordable, portable option that can be used in tandem with remote sensing platforms for a more complete sampling of the ABL.

Diurnal cycles in temperature and humidity characterize ABL transitions, driving changes in ABL height and stability. However, other processes can have additional effects on how the ABL transitions to different states. For example, advection

and subsidence are difficult to quantify, but play important roles in transitions (Angevine et al., 2020). Additionally, clouds can have varying effects on ABL transition periods (Brown et al., 2002). All of which can also affect the ABL height. There are numerous ways to determine the ABL height, many of which are described and tested in Dai et al. (2014) and Dang et al. (2019). Notably, potential temperature proved to be a highly accurate method of estimating ABL height with vertical data

resolution less than 20 m (Dai et al., 2014). Similarly, sharp gradients in humidity have been used to determine the ABL height
for both stable and convective boundary layers when using lidar data (Hennemuth and Lammert, 2006). Dang et al. (2019) also
evaluated different systems to determine ABL height, which did not include RPAS, and determined lidar-based profiles would
benefit NWP. A common thread throughout these studies is that there is no perfect determination for ABL height, nor is there
a perfect platform (Seibert et al., 2000; Dai et al., 2014; Dang et al., 2019). While there will always be benefits and drawbacks
to observation platforms, it is possible that RPAS could marry some of the pros, while reducing costs. Alongside the evolution
of sampling strategies, there follows new ways to determine the ABL height.

Buoyancy is a fundamental force in fluids caused by density differences that can drive vertical acceleration. Buoyant parcels
rise from the warm surface and convectively mix the ABL. This process is the foundation behind most gradient-based ABL
height methods previously mentioned. Angevine et al. (2020) uses a surface buoyancy flux framework to define stages in ABL
transition periods. Additionally, it has been used in attempts to forecast severe weather. Buoyancy is the basis for convective
parameters like convective available potential energy (CAPE) and convective inhibition (CIN). CAPE is buoyancy integrated
between the level of free convection and the equilibrium level, which may not always exist in every environment. In contrast,
CIN is the culmination of negative buoyancy which suppresses thermal lift. Since CAPE is a bulk parameter, the most substan-
tial influence comes at middle troposphere. Climatologically, CAPE has correlated directly with storm intensity (Zhang and
Klein, 2010) but has little short-term prognostic value (Ziegler and Rasmussen, 1998). CAPE and CIN lack the small scale,
near-surface effects needed to understand convection initiation (CI). As a result, mean radiosonde derived values of CAPE and
CIN do not significantly differ between deep convection and fair weather days (Zhang and Klein, 2010). Yet, in the same study,
average low-level (< 5 km) buoyancy does significantly differ. Moreover, single-level, simulated buoyancy values rapidly in-
tensify, overcoming entrainment dilution prior to CI (Houston and Niyogi, 2007; Trier et al., 2014). Additionally, buoyancy is
used to quantify cold pool strength. Simulations indicate that an ample cold pool is key to long-lasting quasi-linear convective
systems (Weisman and Rotunno, 2004). Another facet of buoyancy is its influence on modeled low-level jet (LLJ) speed. The
strength of positive buoyancy had a direct relation with the maximum wind speed for southerly LLJs (Shapiro and Fedorovich,
2009; Shapiro et al., 2016). Conversely, large positive buoyancy impedes the initiation of northerly LLJs, such that negative
buoyancy is beneficial to the northerly LLJs (Gebauer et al., 2017). Proper understanding of LLJs has implications on deep
convection, air quality, and wind energy.

In short, the utilities of buoyancy have been shown by models, yet few studies have substantiated the results with in situ
observations. Recent developments in RPASs allow us to gather the necessary measurements to test these hypotheses in real
environments. Alongside the evolution of observation platforms, there is an opportunity to advance the methods. The time-
height evolution of buoyancy under different ABL phenomena will be analyzed. Using the high spatiotemporal resolution data
from rotary-wing RPAS, ABL features and transitions will be dissected. The following analysis will include two cases: the
diurnal ABL cycle under the influences of an LLJ and pre-convection conditions at two locations within an elevated valley.
The goal is to expound on the unique advantages gained by viewing ABL processes through the lens of buoyancy.

## 2 Campaigns

Here we consider examples of data collected using RPAS and radiosondes during two different field campaigns. Both campaigns aimed to display the usefulness of RPAS under various atmospheric phenomena. The Flux-Capacitor campaign sampled boundary layer transitions under a common Southern Plains occurrence, the LLJ. The Lower Atmospheric Process Studies at Elevation—a Remotely Piloted Aircraft Team Experiment (LAPSE-RATE) campaign was uniquely located at high-altitude with orographically driven circulations and different land surfaces. An aerial map of each location can be found in Fig. 2 of Bell et al. (2020). We will use these data to evaluate the utility of low-level buoyancy in various environments. All flights completed during both campaigns were conducted under Federal Aviation Administration (FAA) certificates of authorization (COA) and overseen by FAA licensed pilots, who were embedded with the research teams. A complete description for each campaign follows.

### 2.1 Flux-Capacitor

The Flux-Capacitor field campaign took place as a test of the 3D Mesonet concept in which a subset of Oklahoma Mesonet stations would include an RPAS capable of regularly profiling the lower troposphere (Chilson et al., 2019). The campaign tested the feasibility of continuous flights to observe the ABL transition over a 24-h period. Flights began at 1501 UTC (1001 LST) 05 October 2018, taking off every 30-min, and with the last flight at 1430 UTC (0930 LST) on 06 October 2018. A flight to 1 km above ground level (AGL) takes roughly 12-min. This campaign sampled the ABL throughout its diurnal cycle and under southerly LLJ conditions. The flight ceiling for Flux-Capacitor was based on line of sight operations up to 1,200 m. Flights took place 28 km southwest of Norman, OK, USA at the Kessler Atmospheric and Ecological Field Station (KAEFS), which is co-located with the Oklahoma Mesonet's Washington station (WASH). Additionally, a radiosonde was released approximately every 3 hours, for a total of 10 soundings. Radiosondes served as a way to validate measurements from RPAS profiles.

### 2.2 LAPSE-RATE

LAPSE-RATE took place in San Luis Valley, Colorado, USA from 14–19 July 2018 (de Boer et al., 2020a). Ten teams gathered to collect atmospheric measurements using RPASs for three targeted missions: CI, drainage flows, and boundary layer transition. Teams distributed across the valley regularly collected synchronized, vertical profiles of the atmospheric state up to 914 m above ground level with rotary-wing RPAS. Additional data were collected using fixed-wing RPAS, radiosondes, and ground-based remote sensors. Although there were many other RPASs and remote sensing platforms used, we will focus on the two Center for Autonomous Sensing and Sampling (CASS) deployed stations. This allows for direct comparisons with the Flux-Capacitor campaign as the same RPAS was used. CASS had three profiling stations. The two main sites were at Moffat School (MOFF) and Saguache Airport (K04V), approximately 27 km northwest of MOFF. To capture the cold air drainage, the team relocated from K04V to Saguache Farms (SAGF) on 19 July 2018, but these data will not be included. The base flight frequency was 30 min, but during an ABL transition, such as pre-convection or drainage flow reversal, the frequency was increased to every 15 min.

This campaign was unique in location and execution. The San Luis Valley has an average elevation of 2,300 m and peaks at 4,000 m above sea level. The valley is arid but contains irrigated cropland creating gradients in temperature and moisture from differing land uses. There is orographic lift, which leads to convection commonly occurring overtop the mountains. Furthermore, mountain-valley circulations affect ABL transitions and air quality. Teams were able to partially sample the valley, which is nearly the size of Connecticut, by completing over 1,200 flights, using 34 different platforms (de Boer et al., 2020b; Pillar-Little et al., 2021). Conditions within the valley were ideal for RPAS flights and observing mesoscale to microscale flow features. de Boer et al. (2020b) provided a description of the weather conditions. In summary, due to limited moisture, temperature and humidity have a strong diurnal cycle that drives flow features. In the afternoon when the ABL is approximately dry-adiabatic, winds are gustier and occasionally enhanced by outflow from mountain convection. The first 2 days of structured flights (15-16 July 2018) were selected to study CI. Both days had moisture advected from the Pacific Ocean with a passing cold front. The 19th was focused on capturing cold-air drainage flow, hence flights began shortly before sunrise.

## 3 Observation platforms

During the LAPSE-RATE campaign, there were numerous RPASs collecting data in addition to remote sensing platforms and ground station observers. All data obtained during LAPSE-RATE can be found at https://zenodo.org/communities/lapse-rate/. Flux-Capacitor utilized the CopterSonde RPAS, radiosondes, and surface observations. In order to have direct comparisons between the two datasets, we chose to only use the data from radiosondes and the CopterSonde. The description of these two platforms follows.

### 3.1 CopterSonde

The RPAS utilized in both field campaigns was the CopterSonde 2. This is a rotary wing quadcopter designed and manufactured by CASS at the University of Oklahoma. The CopterSonde contains three temperature sensors (iMet-XF glass bead thermistors) and three relative humidity sensors (Innovative Sensor Technology HYT 271). These are placed within the shell of the aircraft, protecting them from solar radiation and heat from the motor, which can impact the precision of the measurements (Greene et al., 2018, 2019). Pressure is determined by the MS561 pressure sensor which is built into the autopilot board to aid in altitude control. Built into the shell is an aspirated intake scoop that is designed to consistently draw air across the sensors. It features a sampling technique that adapts to position the intake scoop into the wind. An algorithm using roll, pitch, and yaw details from the autopilot determines the wind speed and direction. Consequently, this improves measurement accuracy and eliminates the need for additional wind speed and direction sensors (Segales et al., 2020; Greene et al., 2019). The sensor scoop was tested in the Oklahoma Climatological Survey calibration laboratory. The bias for each sensor is calculated and applied to the CopterSonde data, which is further described in Segales et al. (2020). These adjustments are built off of trials from previous campaigns such as Environmental Profiling and Initiation of Convection (EPIC; Koch et al. (2018)) and 2018 Innovative Strategies for Observations in the Arctic Atmospheric Boundary Layer (ISOBAR; Kral et al. (2021)).

| Date | Time (UTC) | Number of flights | Avg. flight frequency (min) | Location | Mission |
|---|---|---|---|---|---|
| 20180715 | 1326-1944 | 18 | 15 | MOFF | CI |
| 20180715 | 1400-1915 | 12 | 30 | K04V | CI |
| 20180719 | 1150-1700 | 24 | 15 | MOFF | Drainage flow |
| 20180719 | 1130-1700 | 22 | 30 | SAGF | Drainage flow |
| 20181005 | 1500-2335 | 18 | 30 | KAEFS | LLJ |
| 20181006 | 0000-1431 | 28 | 30 | KAEFS | LLJ |

**Table 1.** Summary of CopterSonde flights from Flux-Capacitor and LAPSE-RATE.

CopterSonde and radiosonde data gathered in these campaigns were compared in Bell et al. (2020), in addition to data from the Collaborative Lower-Atmospheric Mobile Profiling System (CLAMPS) (Wagner et al., 2019). Temperature, humidity, wind speed and direction from each system were compared and showed strong agreement.

### 3.2 Radiosonde

Radiosondes have stood as the standard for atmospheric measurements for over 90 years and have served as a validation tool for many novel sensing platforms. The Vaisala RS92-SGP radiosonde was used for this study. Data from the radiosondes are initiated from ground station data. According to Vaisala technical data, there is a 0.5 °C uncertainty for temperature and 5% uncertainty for relative humidity. The measurement response time for both sensors is less than 0.5 s. Data are recorded and transmitted at 1 Hz. Further information regarding the radiosondes used during LAPSE-RATE can be found in Bell et al. (2021).

## 4 Methods

The CopterSonde allowed for controlled measurements taken at a prescribed frequency specified by the needs of each campaign. Table 1 describes flight strategies in both experiments. Abiding by FAA air regulations, the flight ceiling for Flux-Capacitor was 1,524 m (5,000 ft) above ground level (AGL) with line of sight operations required. Lights affixed to the RPAS allowed visibility into the night, but due to high winds, a majority of Flux-Capacitor flights did not reach the flight ceiling. As for LAPSE-RATE, flights were authorized up to 914 m (3,000 ft) AGL, which most flights reached.

Buoyancy ($\beta$) was calculated at each level using Eq. (1) such that the vertical resolution is the same as all other variables. The parcel's virtual temperature ($T_{v,par}$) was calculated based on parcel theory with the lowest observed temperature and dew point used as the initial inputs. The temperature and relative humidity measured by the RPAS or radiosonde are used to calculate the virtual temperature which functions as the environmental temperature ($T_{v,env}$) while g is the acceleration due to gravity.

$$\beta = g * \frac{T_{v,par} - T_{v,env}}{T_{v,env}} \tag{1}$$

CopterSonde data were recorded at 10 Hz and then downsampled to a 3 m vertical resolution for Flux-Capacitor and 10 m for LAPSE-RATE. As for the radiosondes, the data were vertically interpolated to mimic the sampling resolution of the Copter-Sonde. Example profiles $T_{v,par}$, $T_{v,env}$, and $\beta$ using CopterSonde data are provided in Fig. 1. The detailed account of how the LAPSE-RATE data were processed can be found in Pillar-Little et al. (2021). The summed buoyancy for the radiosondes was calculated up to the flight ceiling for the CopterSonde closest to release time. Since flight ceilings change based on flying conditions, this was done to make the results most comparable.

Two methods are used to determine the ABL height during Flux-Capacitor. The control method finds the height of maximum vertical potential temperature gradient, hereafter called potential temperature method. It was chosen because it has shown to be applicable in stable and convective boundary layers over land (Martucci et al., 2007). The hypothesized method finds the height of the minimum buoyancy gradient, hereafter called buoyancy method. Above a convective boundary layer, there is a capping inversion and the atmosphere becomes more stable; above a stable boundary layer, there is a residual layer which is less stable. Therefore, the height of the ABL should be found where the buoyancy begins to decrease in magnitude sharply. In Fig. 1, this would be approximately where the buoyancy profile intercepts the zero buoyancy line, since the slope is very small. To smooth over some individual spikes in the CopterSonde data, a 5-point (15 m) running mean is applied across the entire profile to derived quantities including, potential temperature, dew point temperature, mixing ratio, and buoyancy. Since these values are not directly observed, the calculations to attain these values may have introduced noise. This is not necessary for the radiosonde data since it was processed by Vaisala software and then additionally vertically interpolated.

## 5    Results

### 5.1    Case 1: Flux-Capacitor

Radiosondes and the CopterSonde were regularly deployed allowing them to be cross-evaluated. Both platforms share similarities in quantities and dimension sampled. Nonetheless, there are stark differences in abilities. Radiosondes are capable of sampling a much higher column, while the CopterSonde's flight ceiling is limited greatly by regulation, technology, and atmospheric conditions. Radiosondes are not true Eulerian profilers; they are advected with the flow, adding quasi-Lagrangian impacts. As a result, observations are coming from downwind of the release site, especially for Flux-Capacitor since there was a strong LLJ. The CopterSonde conducts fixed location profiles delivering true local vertical gradients. Moreover, the cost of a radiosonde profile is much higher than a CopterSonde profile, restricting the temporal resolution of radiosonde releases. Nevertheless, the long-established confidence in radiosondes makes them a validation tool for measurements from the CopterSonde. Figure 2 shows the temperature at the lowest measured elevation from both platforms in addition to the Oklahoma Mesonet's 9 m temperature observation. Given the thermistor response time of $<2$ s, the CopterSonde has enough time to acclimate to the air temperature at 6 m. The radiosondes used in this field campaign take in the station measurements as a boundary condition, which explains the strong agreement between radiosondes and Mesonet data. Initially, the CopterSonde has approximately a $1°C$ warm bias at the lowest elevation. This is a consequence of the shell being heated by the sun during the setup of the site. Continuous aspiration over the sensors above the surface would reduce this effect at higher elevations. The recurrent flights

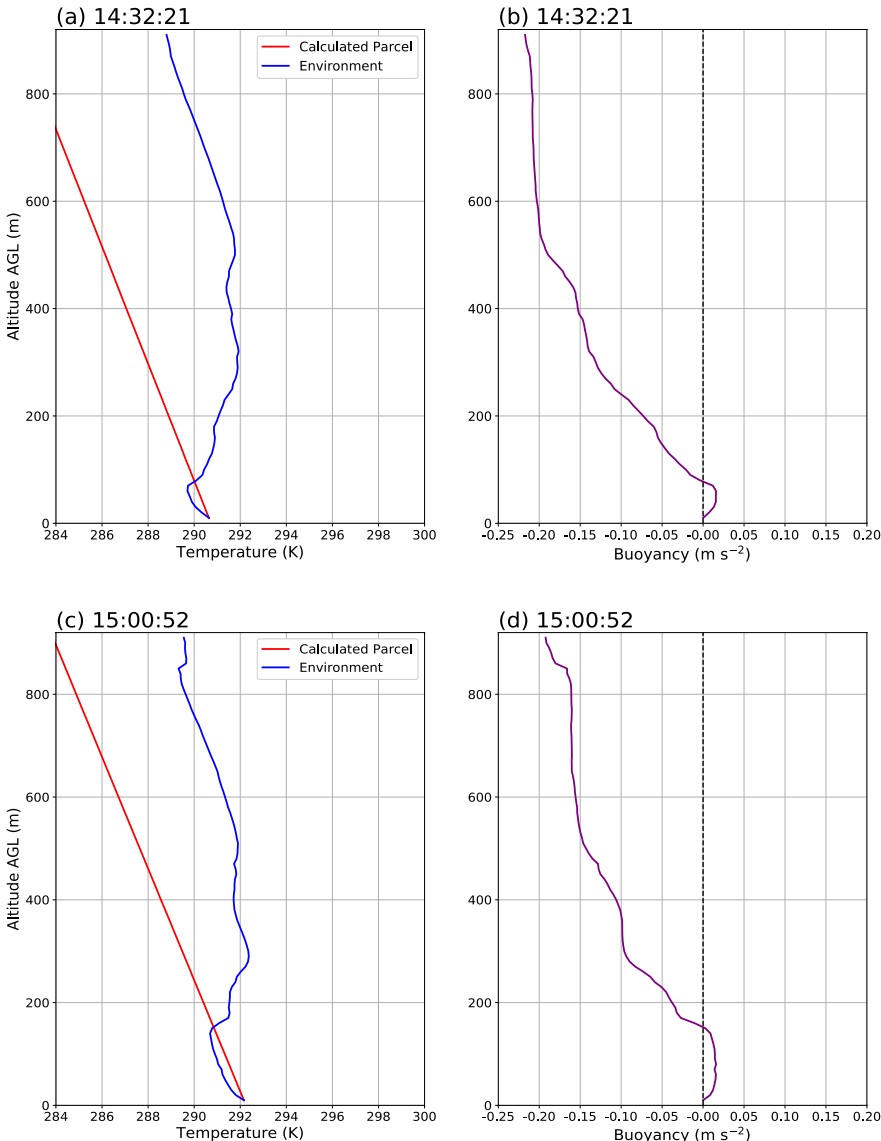

**Figure 1.** Temperature (K) and buoyancy (m s$^{-2}$) profiles from CopterSonde at MOFF on 19 July 2018. (a,c) Temperature observed from CopterSonde (blue) and dry adiabatically lifted parcel (red). (b,d) Buoyancy profile (purple) black dashed line at neutral (zero) buoyancy. All times are in UTC.

following prevented the CopterSonde from sitting in the direct sun long enough to heat up. Such that, the warm bias reduces below 0.5°C after 1607 UTC. Keeping the instrument in the shade until takeoff is now the standard to mitigate this effect. Therefore, there is confidence in the accuracy of temperature measurements, thus the initialization point for buoyancy profiles.

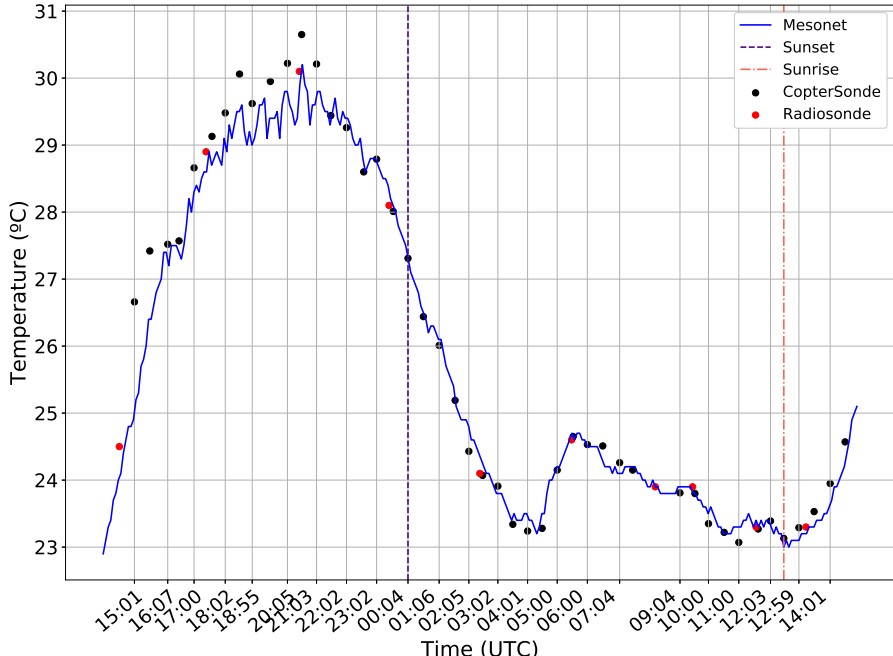

**Figure 2.** Time series of temperature (°C) measured at the lowest level from the Mesonet (9 m) (blue line), CopterSonde (6 m) (black dot), and radiosondes (7 m) (red dot) on 05-06 October 2018. Purple and orange vertical lines represent sunset and sunrise, respectively.

Figure 3 shows the contours of calculated buoyancy in time and height from the CopterSonde (left) and the radiosondes (right) over environmental variables. Profiles from each platform are interpolated over time and height to create the continuous figures. While the interpolation is the same, the radiosonde data are interpolated over 3 h compared to the CopterSonde's 30 min period. Assuming that the environment changes linearly over 3 h is likely inaccurate, especially during ABL transitions such as morning or evening. As a result, the ABL morning transition (1432-1724 UTC) looks much smoother with the radiosonde data (Fig. 3b, d). The atmosphere's turbulent nature is highlighted by fine-scale changes in the wind speed shown by the increased vertical data resolution with the CopterSonde coupled with the increased frequency of profiles (more flights). The change in flight ceilings seen in Fig. 3e shows the limitations of flying in a high wind environment. Nonetheless, the change in buoyancy throughout the time period is similar for both platforms. After 2103 UTC, the surface cools rapidly as insolation decreases and a shallow inversion initiates an stable boundary layer. Buoyancy's rate of change is of interest from 0130 - 0430 UTC when a negative gradient forms preceding the onset of the LLJ. The negative buoyancy is at its peak in an elevated layer where the LLJ forms about 1 h later. The buoyancy gradient aligns with a declining moisture gradient (Fig. 3c, d). This could be attributable to the downwelling of drier, warmer air before the LLJ. Figure 2 shows a local maximum in temperature at the time of maximum wind speed from 400 - 1,200 m. As the warmer air aloft is mixed down, there is a slight rise in buoyancy around 0530 UTC (Fig. 3a, b). The shear instability acts to enhance turbulent mixing and degrade the stable layer to approach a neutral state. Without it, the negative buoyancy would suppress turbulence and lead to greater stratification. The LLJ disrupts the potential

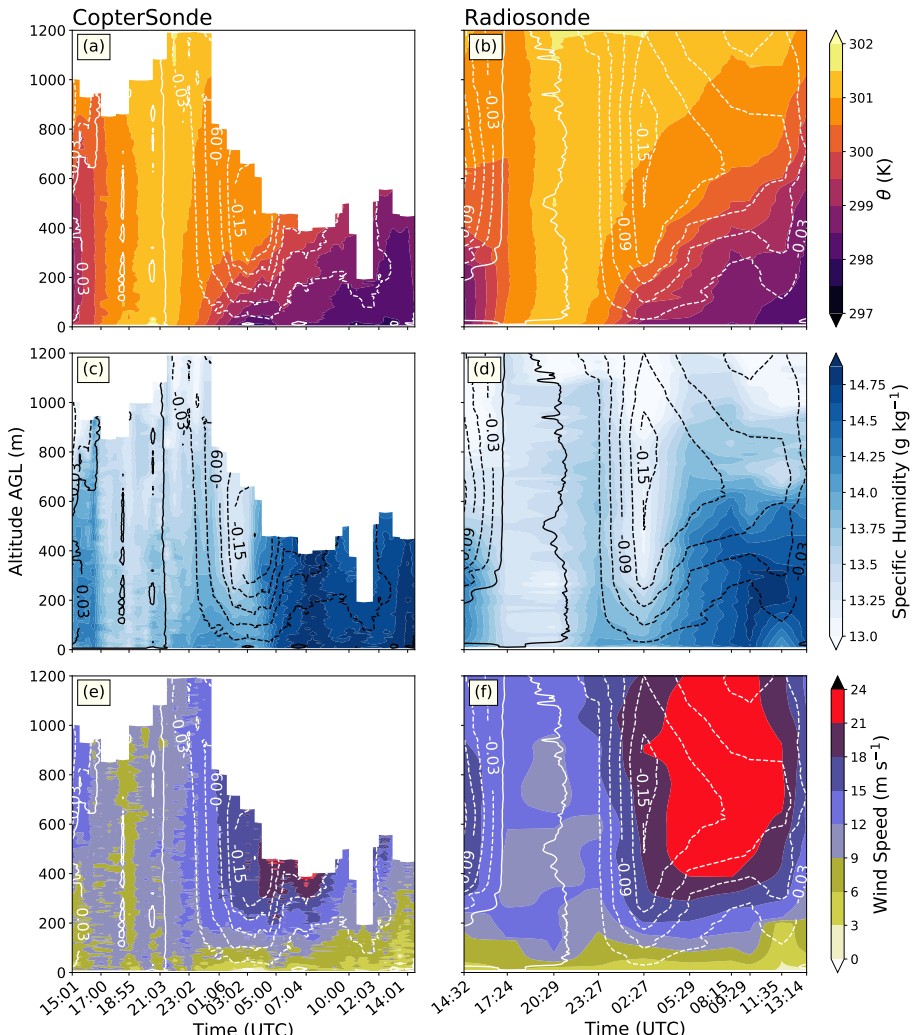

**Figure 3.** 05-06 October 2018 shaded contour fields of (a,b) potential temperature (K), (c,d) specific humidity (g kg $^{-1}$), and (e,f) wind speed (m s $^{-1}$) with buoyancy contours overlain (m s $^{-2}$) at KAEFS. Dashed(solid) contours are negative(positive). (a,c,e) use CopterSonde data and (b,d,f) use radiosonde data.

temperature gradient by redistributing cooler air vertically. Figure 3a shows the surface cooling beneath the jet is not as strong due to mixing. Additionally, there is moisture advected beneath the southerly jet. If the stratification remained, fog may have formed. While the temperature field indicates the formation of an SBL, buoyancy provides more information about the timing and layer which the jet forms. In short, buoyancy helps delineate the interconnections between the LLJ and the ABL.

After the initial analysis seen in Fig. 3, it was observed that buoyancy is roughly constant with height in a CBL. This is expected since buoyancy is a driving force to homogenize the ABL. Therefore, we propose a gradient-based method to find the ABL height from buoyancy profiles. In order to evaluate a new method, the ABL heights determined from the potential

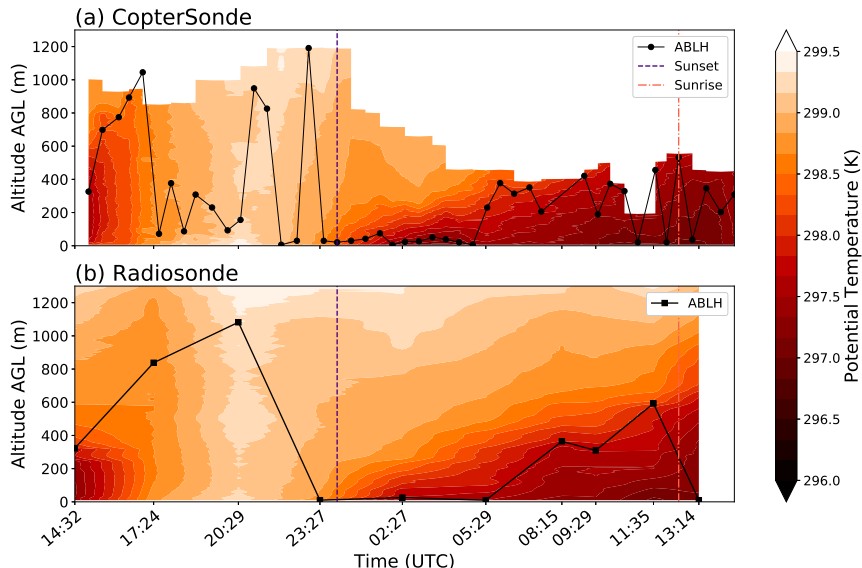

**Figure 4.** Potential temperature field (shaded contours) and black line indicating the ABL height determined by height of maximum potential temperature gradient on 05-06 October 2018 at KAEFS. a) each circle indicates the height determined by an individual flight. b) each square indicates the height determined by an individual radiosonde.

temperature method are also found (Martucci et al., 2007). The potential temperature method, radiosonde derived heights (Fig. 4b) do not change as rapidly as those derived from the CopterSonde data (Fig. 4a). Moreover, the potential temperature method ABL heights from the CopterSonde data are lower from 1724-2300 UTC compared to radiosonde derived heights. It is worth noting that some heights appear to surpass the provided data, this is a smoothing artifact from plotting. The dropoff in ABL height occurs once the mixed layer extends past the flight ceiling. Without a strong transition above the ABL, the potential
temperature method erroneously finds where the surface layer transitions to the mixed layer. The CopterSonde is more likely to find these sharp gradients near the surface because of the increased data resolution at lower levels. Considering most ABL height methods are tested using radiosonde data, it is expected that the potential temperature method works well.

It is worth pointing out the differences in sampled potential temperature. Around 1432-1500 UTC the height of mixed layer disagrees strongly between the two sampling platforms (Fig. 4). The depth of the mixed layer determined by a radiosonde
release is around 330 m (Fig. 4b), while it is around 550 m for the CopterSonde (Fig. 4a). Although, the ABL heights are the same between both platforms. A measurement bias is not suspected; it is likely due to the radiosonde being advected downwind. Figure 3f shows 15-18 m s$^{-1}$ winds in the 350-970 m layer, directly above the ABL top (Fig. 5). Between the 1430 UTC radiosonde release and the first CopterSonde flight at 1501 UTC, the radiosonde would likely be many kilometers downstream of KAEFS. The difference in mixing layer depths can also be seen in the buoyancy data (Fig. 3, 5). The shallow
positive buoyancy region found by the radiosonde leads to a much more negative vertically summed buoyancy value compared to the nearest CopterSonde observation (Fig. 6).

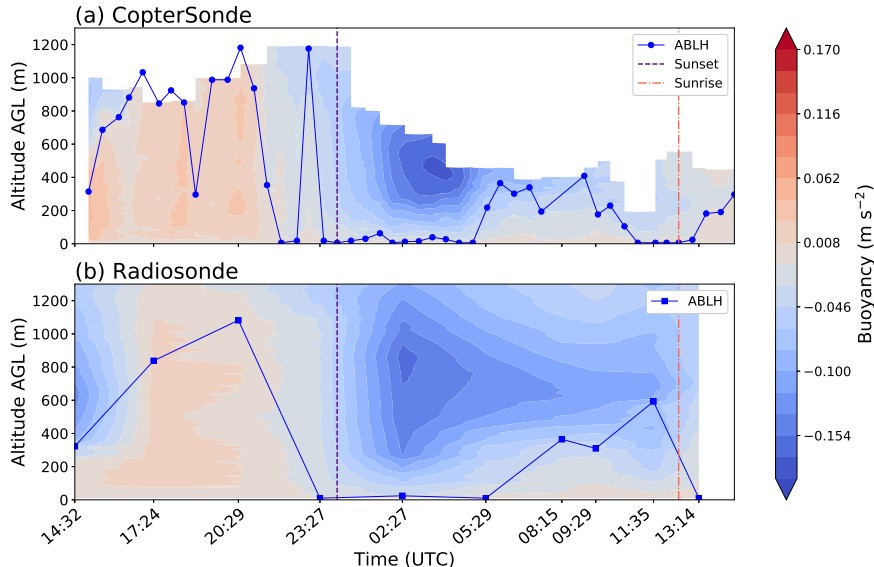

**Figure 5.** Same as Fig. 4 but with buoyancy field and buoyancy-determined ABL heights (blue line).

The proposed buoyancy method is applied to both datasets, seen in Fig. 5. Unlike the CopterSonde derived heights using the potential temperature method (Fig. 5a), the buoyancy method provides more consistent, realistic heights. The 1724-2300 UTC time period has more agreement from profile to profile (Fig. 5a) as well as with the radiosonde derived heights (Fig. 5b).
Furthermore, the radiosonde derived heights using the potential temperature method (Fig. 4b) and buoyancy method (Fig. 5b) are identical throughout the entire period. The correlation (r= 1.0) between the buoyancy method and potential temperature method bolsters confidence that the buoyancy method is promising to determine ABL heights. Once the jet arrives and mechanically mixes the surface layer, there is a rise in ABL heights across all methods and datasets (Fig. 4, 5). Afterwards, the agreement between the potential temperature (Fig. 4a) and buoyancy (Fig. 5a) methods from CopterSonde data improves. As
with most gradient methods, it suggests that the buoyancy method would perform better in convective boundary layers than stable boundary layers.

In a similar manner to the rise and fall of ABL height, Fig. 6 shows the change in vertically summed buoyancy throughout time. Since buoyancy is dictated by temperature differences, diurnal changes in insolation give the graph a sinusoidal shape. This is shown by both platforms even though there is some variability. Peak vertically summed buoyancy occurs at the same
time as peak surface temperature (Fig. 2). Radiative cooling of the surface causes the summed buoyancy to sharply decrease as the sun sets. This agrees with the methods established in Shapiro et al. (2016), the maximum buoyancy occurs a few hours before sunset, in this case 4 hr ahead of sunset. Although, the observed, steady decrease in buoyancy right before the LLJ is over a much deeper layer compared to the modeled results (Fig. 7 from Shapiro et al. (2016)). Upon the arrival of the LLJ, there is a rapid increase in buoyancy, as a result of rising temperature. Turbulent forces act to return the ABL to a neutral state,
allowing the environment to quickly become positively buoyant once daytime heating begins.

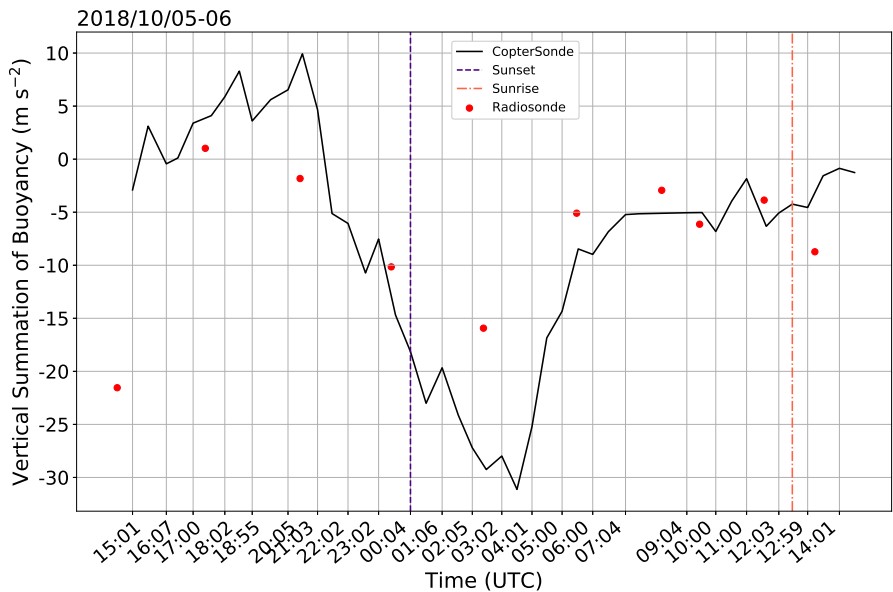

**Figure 6.** Time series of vertically summed buoyancy (m s$^{-2}$) on 05-06 October 2018. Black line is CopterSonde data, red dots are radiosonde data. Purple and orange vertical lines represent sunset and sunrise, respectively.

## 5.2 Case 2: LAPSE-RATE

Before LAPSE-RATE began, forecast models specific to the valley were run to predict which days would be best suited for the different research objectives: CI, drainage flows, and boundary layer transitions. The first 2 days of the campaign (15-16 July 2018) were selected to study CI. Both days had moist environments with a rapidly destabilizing ABL. The weak ridge

and lack of wind shear promoted isolated convection. This study will focus on 15 July 2018 since it experienced CI within the valley, including directly over K04V. Fortunately, convection initiated over a profiling site, thus providing local discrepancies in pre-convection variables.

At 1715 UTC, the automated surface observing system (ASOS) stationed at K04V reports distant lightning and archived radar shows convection 25 km north of the site. Two hours later, the same ASOS station reports a thunderstorm. As a result,

flights at K04V end 30 min before flights at MOFF. At this time, archived radar shows the deepest convection is still 10 km north (Fig. 7). This highlights the issues of radar coverage within the valley. There is a delay in ASOS storm report and radar storm visibility because storms are not seen by the radar until they are taller than the mountains. Around 1955 UTC, CI occurs 4 km east of MOFF. Outflow winds hit MOFF during the last flight at 1944 (Fig. 8e). At 2001 UTC, the site only receives light rain with stronger convection moving north. These times will become useful as we analyze the buoyancy and moisture fields.

Figure 8c and 8d show how buoyancy and moisture evolve in time with height at both sites. In the morning, there is more moisture throughout the column at MOFF than K04V. The location of MOFF at the base of the valley leads to more moisture accumulation than within a sloped canyon. At K04V, the drier air near the surface heats more quickly and leads to faster

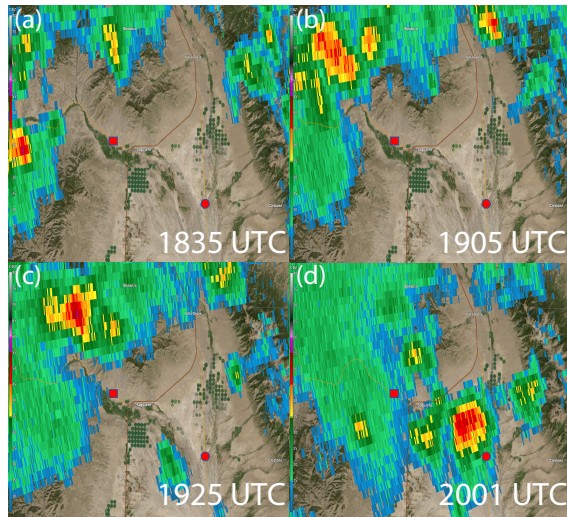

**Figure 7.** 0.5° level reflectivity (dBZ) from the KPUX radar in Pueblo, CO over the San Luis Valley on 15 July 2018. Red square indicates the K04V site and red circle indicates MOFF. a) 1835 UTC (1235 MDT) b) 1905 UTC (1305 MDT) c) 1925 UTC (1325 MDT) d) 2001 UTC (1401 MDT)

destabilization (Fig. 8b). From 1700-1740 UTC there is a strong positive buoyancy gradient in time at K04V. Here, buoyancy is uniform throughout the entire layer. In addition, low-level moisture increases with time. It is possible this is a result of moisture convergence induced by outflow from the storms to the north. The strong positive buoyancy aids in vertical transport of moisture. Rapid destabilization coupled with deepening low-level moisture creates a favorable convective environment. Consequently, the K04V ASOS reports 14 m s$^{-1}$ gusts from 1954-2030 UTC. Conversely, at MOFF there is little change in buoyancy with time and moisture decreases with time (Fig. 8c). Although at 1700 UTC, there is vertical transport of moisture accompanied by a layer of increased buoyancy in the lowest 500 m (Fig. 8c). While there appears to be parcel ascent, it was not enough to initiate convection. Once convection begins in the valley (2000 UTC), MOFF is drier and neutrally buoyant. As a result, the storm favors northward propagation, away from MOFF.

Even though the two sites are only about 27 km apart, there is a difference in how buoyancy evolves in time, showing its spatial sensitivity. Variations in moisture over the two locations change the rate of surface heating. Differences in moisture could be attributed to different land cover or different positions within the valley. Surface conditions influence parcel trajectory, and buoyancy infers deviations about the environmental profile. Increased representation of land-air interactions is a valuable asset to any forecasting tool. Unlike potential temperature, buoyancy is directly impacted by surface conditions at each level of calculation. Buoyancy aligns with changes in moisture where potential temperature does not (Fig. 8c). Such that microscale features can be recognized more readily using buoyancy. Whereas in Fig. 8c and 8d, the temperature field at each site is overall very similar. Except at 1930 UTC, the MOFF site has a cooling throughout the layer by 1°C, likely from the outflow boundary, which the K04V site does not experience (Fig. 7c and 8b). At the time of the rapid increase in buoyancy, the surface temperature at K04V is about 3°C warmer than at MOFF. A shift in surface temperature by a few degrees may be overlooked, but buoyancy

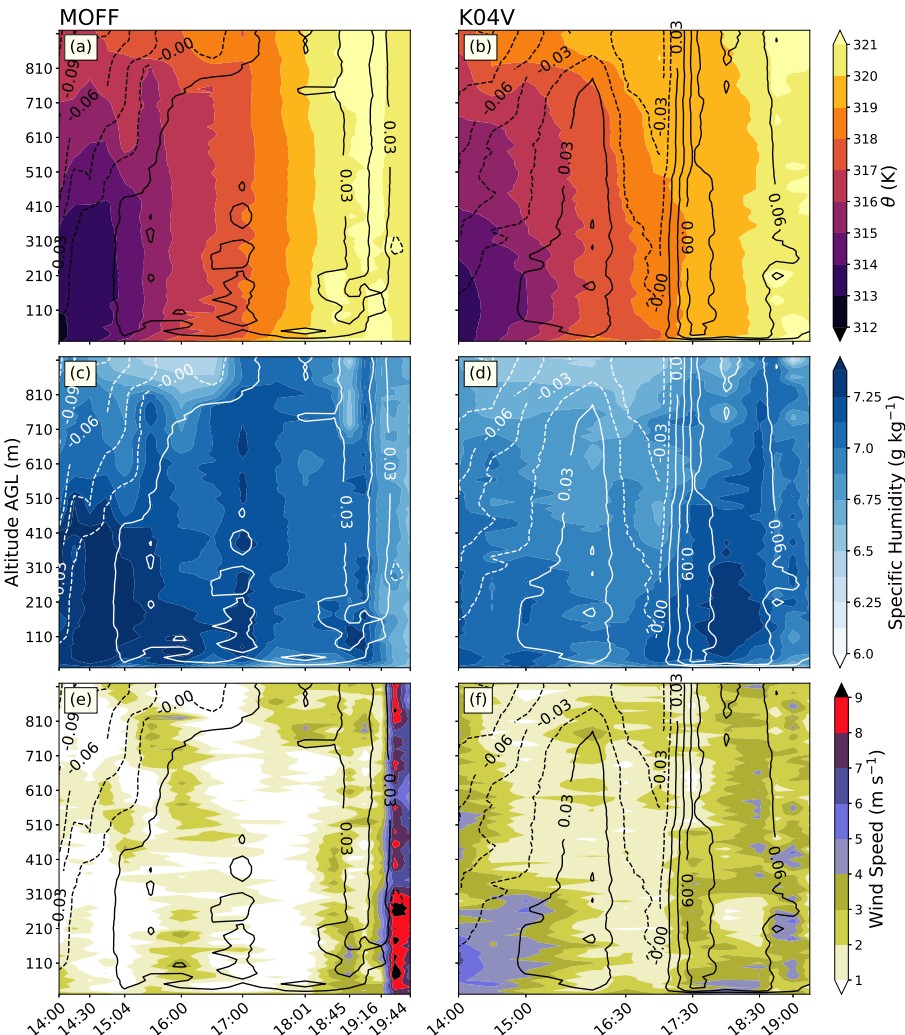

**Figure 8.** Same as Fig. 3 but on 15 July 2018 and (a,c,e) are at the MOFF site and (b,d,f) are at the K04V site.

accents how that affects the column. Overall, buoyancy in convective settings is sensitive to environmental changes, which may predate the amplification or weakening of convection.

In contrast, Fig. 9a focuses on buoyancy and moisture for a non-CI case that experienced cold air drainage. Drainage flows
310    are driven by radiative cooling causing the air above the surface to become denser and descend the valley walls. The process continues throughout the night resulting in a cold pool of air at the base of the valley. Thus, a strong temperature inversion settles in with subsequent negative buoyancy. The strong easterly winds throughout the early morning confirms strong downsloping flow (Fig. 9b). Figure 9a quantifies the intensity of stability. Until 1400 UTC, the stable boundary layer has clearly stratified layers up to 300 m. There is some variability in the vertical extent from 1133-1215 UTC, which could be caused by turbulence.
315    It is of note that the region of stratification aligns with a layer of moisture. Such that the specific humidity decreases and

becomes homogeneous beyond the same height that the buoyancy gradient decreases. Above that level (300 m) and below the stagnation level, there is homogeneous easterly flow (Fig. 9b). All of which suggests a transition to the residual layer. About an hour after sunrise (1245 UTC), the surface has warmed enough to dilute the density current, which causes the flow to slow within the lowest 100 m (Fig. 9b). Thereafter the surface warms, and a shallow mixed layer grows, but it is still capped by a stable layer. This region is convectively neutral with increasing mixed layer depth (Fig. 9a). In the absence of moisture advection, the specific humidity illustrates the vertical mixing. Not until 1530 UTC is the surface warm enough to initiate the southerly up-valley flow (Fig. 9). This boundary layer transition is unlike either of the other cases examined. Since cold air pools into a thick layer at the base of the valley, the magnitude of stability surpasses what would occur from radiative cooling over flat land. As a result, valley cold pools can lead to persistent fog (Chachere and Pu, 2016). Understanding the timing of mechanical mixing and fog dissipation would aid aviation forecasts.

## 6  Conclusions

This study uses buoyancy measured from RPAS to describe transitions within the ABL. Two cases are evaluated to understand the versatility of using low-level buoyancy. Recent developments in weather sensing RPASs allow for high-frequency sampling within the ABL. The CopterSonde measures at a higher vertical resolution than radiosondes with comparable accuracy (Fig. 2, 6). Buoyancy has been used prevalently in model studies to interpret microscale to mesoscale processes. The spatial and temporal sensitivity of buoyancy allows for a more detailed interpretation of fine-scale processes. The results show that buoyancy has a diurnal cycle coinciding with the solar cycle (Fig. 6). Buoyancy, like potential temperature, reflects the mixed-layer below the capping inversion, such that the maximum height of constant buoyancy appears to transition with the ABL height (Fig. 3, 8).

The application of vertical buoyancy gradients to derive the ABL height shows promising initial results. The potential temperature and buoyancy method derived heights perfectly correlate (r= 1.0) when radiosonde data are used. The correlation from the two methods, using CopterSonde data is not as strong (r= 0.45). Although, the buoyancy method heights are comparable to the heights derived from the radiosonde data. These heights agree better across methodologies with the radiosonde data than the potential temperature method with different data sources. Inherently, more cases need to be evaluated to have full confidence in the method. Moreover, there are avenues to improve the buoyancy method, particularly when using the CopterSonde data. The high vertical resolution leads to noise in the profiles that is erroneously picked up as the ABL height. Logical arguments will be applied to reduce inconsistencies in ABL heights for consecutive flights. Also, a technique to exclude heights that are beyond the flight ceiling is necessary. The success of this simple method gives credence that with proper improvements it could become a trusted ABL height definition.

The two cases show the versatility of buoyancy in different scenarios. While it is beyond the scope of this study to interpret the factors creating the buoyancy gradient, they agree with past findings. Figure 3e and 3f displays a negative buoyancy gradient beginning two hours before the jet arrives. It is expected that there is a stable boundary layer during nocturnal LLJ (Blackadar, 1957). Moreover, sinking air ahead of the jet would increase stability. As for case 2, Fig. 8b shows an acceleration in positive buoyancy leading up to CI. This agrees with Trier et al. (2014) that there is rapid destabilization before CI quantified by

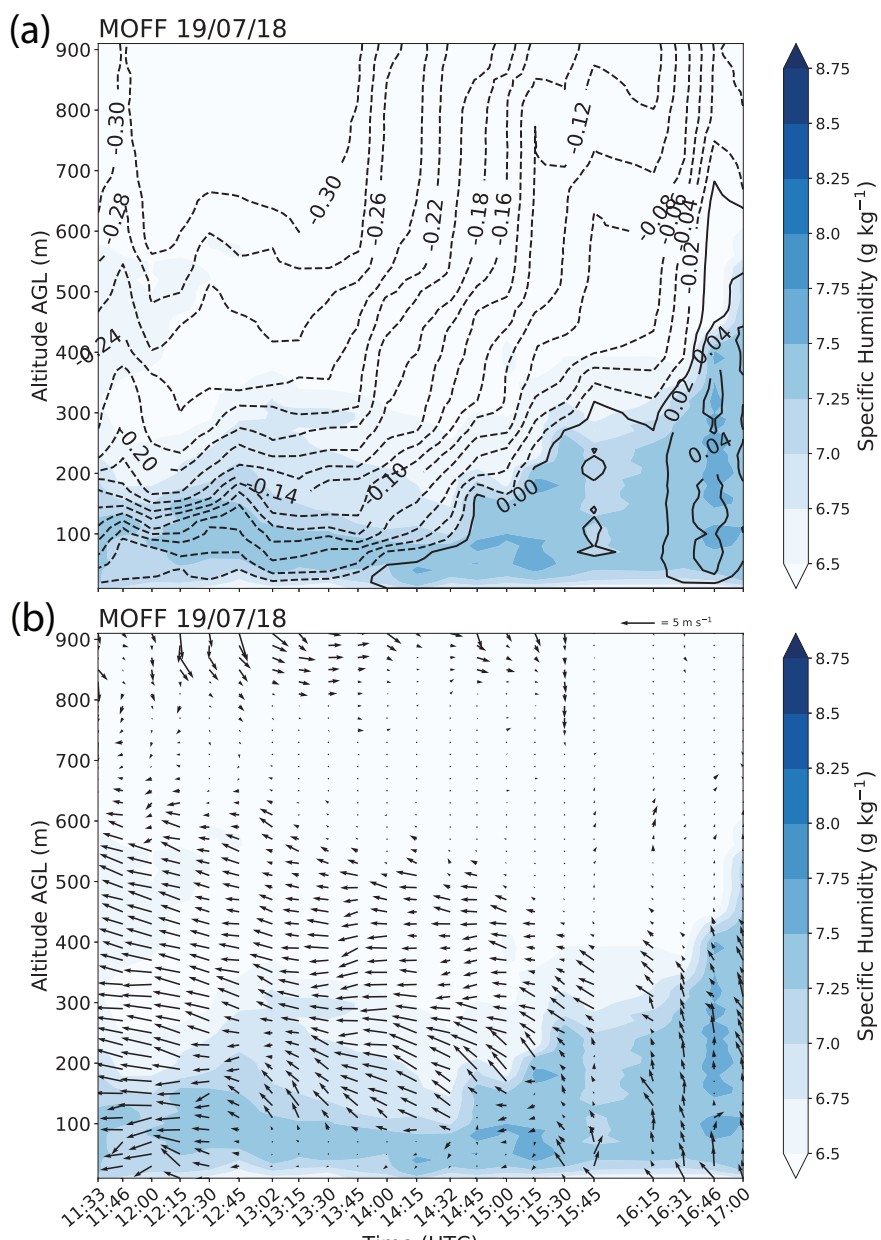

**Figure 9.** (a) Buoyancy contours ($m\,s^{-2}$) where solid(dashed) are positive(negative) and filled contours are specific humidity ($g\,kg^{-1}$). (b) Wind vectors over specific humidity ($g\,kg^{-1}$) filled contours from CopterSondes on 19 July 2018 at MOFF.

parcel buoyancy. MOFF has weaker buoyancy (Fig. 8a) and results in shallower convection than K04V. Further investigation
350  is required to see if this is a common occurrence preceding other events.

Real time sampling allows for data assimilation and improvement in ABL representation for NWP. There is potential for buoyancy to evaluate many other microscale to mesoscale processes. Measurements taken along a dryline could increase understanding of where horizontal convective rolls (HCR) occur. HCR are conducive for CI due to vertical motion and thermodynamic gradients described in Weckwerth et al. (1999). Furthermore, understanding cold pool strength and propagation would aid in forecasting severe weather like mesoscale convective systems. Additionally, RPASs can fill a gap in measurements in urban settings to increase understanding of turbulence and aerosol transport. Buoyant plumes transport aerosols throughout the city, but measuring this typically requires non-permanent towers and tracers, such as smoke or colored aerosols. RPASs deployed to determine buoyancy within the urban canopy could help improve air quality predictions. The applications for low-level buoyancy go beyond the topics evaluated in this study. Meanwhile, RPASs can collect measurements during processes that are not easily accessible. Together they can help address processes not adequately realized.

Regular RPAS profiling opens avenues for increased data assimilation for climate, air quality, and NWP. Buoyancy is just one variable that has shown use in describing the state of the ABL, which was not previously accessible. Buoyancy is sensitive, physical, and simple. Furthermore, remote sensing platforms could use this technique. This study is a simple starting point to revive a classically defined variable in light of new technology. Buoyancy measured by RPASs can describe ABL transitions with little computational power while providing more information than traditional ABL variables.

*Data availability.* Data from the LAPSE-RATE campaign are openly available at DOI:10.5281/zenodo.3737087

*Author contributions.* Data was collected by T.B., E.P.L and P.C.; Data processing and quality control were conducted by T.B.; Data analysis and visualization was performed by F.L.; The manuscript was written by F.L. with contributions from all co-authors

*Competing interests.* The authors declare that they have no conflicts of interest.

*Acknowledgements.* Funding is provided by the NASA University Leadership Initiative under Grant Number 80NSSC20M0162. This research has been supported in part by the National Science Foundation under Grant Number 1539070 and internal funding from the University of Oklahoma Office of the Vice President for Research and Partnerships. The authors would also like to thank the greater San Luis Valley community for welcoming the LAPSE-RATE campaign and providing land access for flight operations. Additionally, we thank those CASS members who gathered and processed the CopterSonde data for both campaigns.

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
