# Peer review of "Low-level buoyancy as a tool to understand boundary layer transitions"

_Atmospheric Measurement Techniques, 2021_

## Author Comment (AC1)

Francesca M. Lappin
Tyler M. Bell
Elizabeth A. Pillar-Little
Phillip B. Chilson

**Authors' Response to Reviews of**

**Low-level buoyancy as a tool to understand boundary layer transitions**

*Atmospheric Measurement Techniques,* 2021-68
* * *
**RC:** *Reviewers' Comment*,     AR: Authors' Response,     ☐ Manuscript Text

**1.  Reviewer #2**

**1.1.  General Comments**

RC:  *The manuscript describes a method of using buoyancy in the ABL (in particular relatively close to the surface) as diagnostic parameter for process studies in ABL research and applies this method on rather limited data sets from RPAS and radio-soundings gained during two field campaigns (Flux-Capacitor and LAPSE-RATE). Although I clearly see the potential of the proposed method, I would like to be convinced a bit more on the usefulness by also providing some of the classical temperature/potential temperature plots as direct comparison with the more common way of looking into boundary layer development (that could e.g. be done by adding plots with the contours of buoyancy over the temperature/potential temperature field for figures 3, 8 and 9). Having in mind that there might be a radiation error in the temperature (and potentially also the humidity readings), in particular in the beginning of a flight during daytime, it might also be considered to use the observations of temperature and humidity from a fast installed and ventilated sensor at the ground to determine Tpar in equation 1, instead of the value taken from the lowest level of the CopterSonde. The language is not always scientifically concise and leaves some room for mis-interpretation/mis-understanding. I have stated some of the occurrences in the specific comment section, but I also suggest another round on that by one of the more senior co-authors of the manuscript. The figures are also not yet in a high-quality state for final publishing. Here I recommend (in addition to further comments in the specific section below) to consider the following changes: I suggest to remove the figure titles that are given in many of the sub-plots; this information is partly given in the axis text and in the figure caption I am not sure if the numbering/charactering of the figures in capital letters is the correct format for BLM The axis numbers/labels and legends/legend labels are often too small and hard to read I clearly see a potential of the proposed method, the manuscript will, however, require certain major revisions described above before qualifying for publication in AMT.*

AR:  I have added plots of potential temperature, specific humidity, and wind speed for Flux-Capacitor and the CI case. Additional analysis was included to draw connections between buoyancy and the other state variables. All figures were improved to reduced title verbiage and align with BLM numbering.

**2. Specific Comments**

**RC:** *Line 16: sounds like only warm air is transported vertically, I suggest removing "warm" here*

AR: "warm" was removed

**RC:** *Line 20: I feel a bit complicated sentence structure at the end; could "are based off of decades old observation methods" be replaced by "are based on observation methods that are decades old"*

AR: suggested replacement accepted

**RC:** *Line 58: "Buoyancy is a fundamental principle in fluids caused by density or temperature differences" one example for those scientific inconcise/incomplete statements; buoyancy is related to density, period! It's temperature and humidity that effect density! So mixing density and temperature as done in this sentence should be avoided.*

AR: "or temperature" was deleted

**RC:** *Line 75: I feel there is a verb missing: "…but few studies have yet to substantiate…"*

AR: The line has been rewritten as "..yet few studies have substantiated the results with in situ observations."

**RC:** *Line 121: CASS has already be used as abbreviation above, so no need to introduce it here again; but please check if it was introduced properly on first appearance*

AR: The spelled out abbreviation was removed from line 120 and added to its first mention in line 104.

**RC:** *Line 131: the more appropriate reference here might be Kral et al., 2020: Kral, S. T., J. Reuder, T. Vihma, I. Suomi, K. Flacké Haualand, G. H. Urbancic, B. Greene, G.-J. Steeneveld, T. Lorenz, B. Maronga, M. O. Jonassen, H. Ajosenpää, L. Båserud, P. B. Chilson, A. A. M. Holtslag, A. Jenkins, R. Kouznetsov, S. Mayer, E. A. Pillar-Little, A. Rautenberg, J. Schwenkel, A. W. Seidl, and B. Wrenger, Innovative Strategies for Observations in the Arctic Atmospheric Boundary Layer Project (ISOBAR) — Unique fine-scale observations under stable and very stable conditions. Bulletin of the American Meteorological Society, Early Online Release, doi:10.1175/BAMS-D-19-0212.1, 2020*

AR: The 2018 ISOBAR citation was replaced with the suggested 2021 publication which provides a better description of RPAS involvement.

**RC:** *Line 149/150: "with the lowest observed temperature and dew point used", this might be misleading and is at least unclear; do you mean (as I assume) the temperature and dew point at the lowest elevation of your flight, or do you mean the lowest absolute value of temperature and dew point of your profile?*

AR: The sentence was restructured to explain that the observation comes from the lowest elevation.

**RC:** *Caption figure 1: for a) and c) are the colors messed up! CopterSonde should be blue and Parcel red!*

AR: The mistake was remedied to accurately portray the plots and agree with the legend.

**RC:** *Figure 1, profiles for 15:15 UTC (lower panels): I assume the lowest 200 m indicate the mixed layer of the developing CBL after sunrise. I am a bit puzzled over the super-adiabatic slope of the environmental curve over the whole BL/ML depth. I can't explain this other than by a systematic bias from your measurements, e.g due to sensor time lag!*

AR: The superadiabatic layer only extends about 50 m. The remainder of the layer is dry adiabatic. Also, a

systematic sensor bias would have implications throughout the entire profile causing a linearly propagating error. Radiosonde data also supports a shallow superadiabatic layer during the day (attatch fig).

**RC:** *Line 167: There should be an "and" before "then". Or did I mis-understand the meaning of this sentence*

AR: 'and' added prior to 'then'

**RC:** *Line 175: replace "greater" by "higher"*

AR: replaced "greater" by "higher"

**RC:** *Line 176: replace "radiosondes" by "radiosonde ascents" or "radiosonde releases"*

AR: 'releases' added to follow radiosondes

**RC:** *Figure 2: legend by far too small font size! I also suggest to plot the CopterSonde data also only as dots (as for the radiosondes) and not as line; the time labels should be limited to hours:minutes and the font should be increased; this last comment applies for all plots with corresponding time axis labelling*

AR: All figures will be updated to reflect the suggestions made my the referee.

**RC:** *Line 180/181: "This is a consequence of the shell being heated by the sun before takeoff." Could this be an explanation for the superadiabatic slope observed in figure 1; the enclosure leads to an overheating in the beginning of ca. 1 deg C that is then slowly reduced due to ventilation during ascent; would perfectly explain an artificial super-adiabatic slope.*

AR: The phrasing of when shell heating was impactful has been altered throughout lines 177-184. The 1 deg error is in the morning and not during the strongest insolation, but rather during the time the CopterSonde was sitting in the sun the longest. During the sunniest part of the day, there are some clouds but the difference in temperature readings is roughly 0.4 ºC.

**RC:** *Line 182: "Moreover, the warm bias reduces to near-zero at 2103 UTC." For me this is already the case for 18:30! It would be nice to have the time series of global irradiance to see e.g., the effects of cloudiness in the afternoon reducing/avoiding this warm bias.*

AR: Figure 1 does not suggest that there is a correlation between irradiance and the warm bias. The timing of when the bias is reduced has been changed from 2103 to 1607 UTC in the text.

**RC:** *Line 199/200: this sentence is very hard to read, I suggest to replace "as potential temperature method, CopterSonde derived heights" by "as those derived from the CopterSonde data.*

AR: The sentence is reworded as the referee recommended.

**RC:** *Line 203/204: "The CopterSonde is more likely to find these sharp gradients near the surface because of the increased data resolution at lower levels." However, the CopterSonde could also pick up artificial gradients here, just due to the fact that the potentially overheated sensor compartment slowly adapts to real ambient temperatures during flight.*

AR: While possible, the authors find it unlikely that the CopterSonde picked up on artificial gradients. The CopterSonde's higher sampling rate and slower ascent rate collects more data than a radiosonde. The duplicate sensors also reduce the likelihood of all three sensors being overheated and lagged. It seems more likely that the radiosonde ascended too quickly to observe microscale changes to potential temperature since the data collection rate is 1 Hz and ascent rate of about 5 m s$^{-1}$. Furthermore, the CopterSonde is gathering measurements on a parcel scale. It is possible that the gradients are implications of measurements

transecting rising thermals. The new paradigm of these types of measurements open up possibilities to capture traditionally missed microscale features.

RC: *Line 259/260: the statement "the potential temperature is independent of the surface conditions" is at least highly disputeable! Maybe you should specify in more detail what you wnt to achieve with this sentence.*

AR: This statement was rephrased to "Unlike potential temperature, buoyancy is directly impacted by surface conditions at each level of calculation. Such that microscale features can be recognized more readily using buoyancy." (lines 264-265).

RC: *Lines 263 and 272: replace "Radiational" by "Radiative"*

AR: Both 'radiational' uses are replaced by 'radiative'

RC: *Line 267/268: "Figure 9 shows how deep the stable layer is until 1400 UTC."; How deep would you diagnose the SBL at that time? At around 600 m where the buoyancy gradient seems to disappear, or somewhere between 200 and 400 m, where the gradient becomes distinctly weaker? I miss a bit more explanation how the buoyancy concept can be used as diagnostic tool for SBL height.*

AR: 'Deep' was supposed to be interpreted as a synonym for intense and not a measure of the ABL height. The lines have been rewritten to remove the confusion. Lines 266-271 were added to remark on features which show potential to mark the depth of the SBL based on Figures 9 and 10. The paragraph was also restructured to flow better.

RC: *Line 269/270: "Not until 1530 does the wind direction shift to southerly, generating some positive buoyancy (Fig. 9, 10)."; This sounds like the wind direction shift is causing the change in buoyancy, I would argue that the change in wind direction is a result of the processes (e.g., differential surface heating) that affect buoyancy; maybe worth of rephrasing this sentence.*

AR: Lines 282-283 were altered to better explain the cause of the southerly wind.

RC: *Line 279: replace "more" by "a more detailed" or "an in depth"*

AR: 'a more detailed' was added.

RC: *Line 280/281: "below the ABL", that should either read "in the ABL" or even in the "mixed layer of the ABL" or "below the capping inversion"*

AR: 'below the capping inversion' was substituted for 'below the ABL.'

RC: *Line 306/307: "There are countless applications for RPASs to sample previously difficult aspects of the atmosphere." This is a very oversimplified formulation, please write a bit more detailed what you mean, e.g. something along the lines "to collect atmospheric measurements for processes that are still not adequately understood"*

AR: Lines 307-309 were rewritten to summarize the paragraph more specifically than the previous version.

RC: *General comment for the references; check citation of pages, sometimes with pp.! mostly without; so please homogenize*

AR: All citations were checked and the journals are all capitalized. Some citations were updated from Early Release to published which remedied the occasional 'pp.' accompanying the page numbers.

[Figure]

Figure 1: Time series of solar irradiance (W m$^{-2}$, brown line), Mesonet 9 m temperature (ºC, blue line), CopterSonde temperature (ºC, black dot), and radiosonde temperature (ºC, red dot) on 05-06 October 2018
.

Line 330: journal name should be upper case

Line 333: reference Båserud is incomplete (journal name, volume, pages etc)

Line 345: journal name should be upper case

Line 354: journal name should be upper case

Line 360: journal name should be upper case

Line 373: journal name should be upper case

Line 396: journal name should be upper case

Line 406: journal name should be upper case

Line 407: journal name should be upper case

Line 419: journal name shouldn't been abbreviated

Line 429: journal name should be upper case

Line 434: journal name should be upper case

Line 439: journal name should be upper case

Line 457/458: journal name should be upper case

Line 464: journal name should be upper case

---

## Author Comment (AC2)

Author Responses are enclosed in square brackets

General Comments:

I find the manuscript to be well-written. The authors demonstrate how observations of buoyancy from remotely piloted aircraft systems operating in the ABL can have practical applications in the diagnosis of ABL height, convective initiation and low-level jets. The authors utilize two field studies for evaluation, both of which are appropriate for the kind of (free-convection-driven) environments where buoyancy measurements are likely to be useful.  Specifically, the authors demonstrate how the proposed buoyancy-based method for ABL height diagnosis mitigates misdiagnosis from noise in the UAS-measured fields and is a viable method for both UAS and radiosonde-based profiles. One could argue that this article is less about RPAS observations and more about buoyancy as an ABL diagnostic, which would make it better suited for a journal other than AMT. I think the authors' well-written introduction and conclusion sections provide enough context for this manuscript to stand as an RPAS application, however.

I have listed below some specific comments regarding some of the statements made and the figures. I think some non-trivial adjustments are needed in places, but the foundation of the investigation is sound and can be revised into a manuscript ready for publication.

Specific Comments:

L 48: The authors characterize "ABL transitions" on this line through its impact on ABL height and stability and proceed with a strong overview of ABL height estimation methods. I think the description of ABL transitions could be expanded a bit beyond this one line, given the history of this topic itself and its prominence in the manuscript title, to help frame the term for the reader. For example, when is the temporal evolution of an ABL a 'transition' and when is it not? I agree with the authors' general characterization here; my only suggestion would be to include the influences from the 3-D flow on ABL transition that may not be diurnal, e.g., internal boundary layers, mesoscale circulations (such as the authors' example later around L 270?), etc., and transitions in forced convection environments). It's also worth noting that the word transition is used later (L 202) in a different context, seemingly referring to changes in the vertical temperature profile slope, rather than changes in ABL 'regime', which is the more common usage for the 'transition' term, I would argue. My impression after reading the full manuscript is that the authors are using low-level buoyancy from RPAS observations to diagnose boundary layer features and characteristics.

[The authors added L 54-57 were added along with two new citations (Angevine et al. 2020 and Brown et al. 2002) to add context to the term 'transition.' The Angevine paper has an

in-depth discussion on ABL transitions and the paraphrasing added to this manuscript conveys to the various impacts on ABL transitions. The L 202 use of 'transition' was changed to 'vertical potential temperature gradient.']

L 128-132: Were the sensor bias corrections applied here identical a particular past campaign, or were new values used? If the latter, what were the new values and how were the corrections applied?

[The bias corrections are identical to the methods described in Segales et al. 2020 (AMT), regardless of the campaign. Each system has a designated scoop and sensor package that has a unique set of offsets gained from calibration upon fabrication. I added a direct point to this citation in lines 129-130 to make that clearer. Also, the scoops are calibrated periodically for sensor drift.]

L 173-174: Minor point, but I would disagree that radiosondes are Lagrangian tracers. The radiosondes here are balloon-launched, which provide lift that enforces upward motion (initially), regardless of the ambient vertical velocity (whether upward or downward). A tracer follows the ambient fluid velocity in three dimensions.

[The authors completely agree. This was brought to our attention after submission. Lines 175-176 were changed to reflect that radiosondes are more quasi-Lagrangian tracers.]

L 193-194: This sentence seems a bit unnecessary and I would suggest that it be omitted. Buoyancy is sensitive to temperature difference between a parcel and its environment, regardless of whether the parcel is in the lower troposphere.

[The sentence was omitted.]

L 195: The sentence reads: " . . . buoyancy becomes convectively well-mixed .. . ." Do the authors mean to say that the ABL becomes convectively well-mixed? Buoyancy being a force, I'm unclear on what is meant by buoyancy being convectively well-mixed. Following the authors' definition in (1), in a well-mixed unsaturated environment, the environmental temperature would follow the adiabatic lapse rate, which a buoyant unsaturated parcel would also follow. It's arguably the components of the expression that are "well mixed" (or constant). Perhaps an alternative expression would be to call it invariant? The expression is also used around L 280 where the authors first introduce "isotropic buoyancy". I think this term is closer to the conditions described, though isotropy does carry a connotation of direction.

["Convectively well-mixed was changed to "roughly constant with height in a CBL." The term "isotropic" was changed to "constant" to not convey that buoyancy is uniform in all directions, per the turbulence connotation. Also, in L 280 the structure was changed to reflect that there are parallels in vertical structure between potential temperature and buoyancy, but they are not both mechanically mixed.]

L 202-203: As the authors insinuate, this example shown in Figure 4 is not appropriate for the potential temperature method applied to these specific CopterSonde data due the deep well-mixed layer. Despite that, have the authors experimented with applying some kind of low-pass filter to the CopterSonde measurements to truncate the high-frequency variability? For this example, I have to wonder if, imagining the CopterSonde flew higher or the mixed layer were shallow enough for the CopterSonde to capture it fully, would the method applied to the CopterSonde data in Figure 4 still yield erroneously low ABL heights?

[The authors tried a handful of methods to eliminate the heights being found erroneously low. A low-pass filter was applied, as were linear splines to estimate gradients, and logical statements to not allow heights to be found lower than 200~m. None of these provided desired results, even the last option just moved the height to exactly one data point above the height restriction. It is because within the mixed layer the vertical gradients are miniscule, so the only sizable gradient occurs near the top of the surface layer. From 1432-1700 UTC, we can see in Figure 4 that the mixed layer is within the flight ceiling. As a result, the ABL height grows logically and agrees well with the radiosonde data, which suggests that it does work if the entire mixed layer is captured. Unfortunately, in the US it is difficult to get airspace cleared much higher than 1200 m. While the authors believe that if we could fly higher we would have better success, it is difficult to prove it.]

L 210-212: The authors contend that local winds combined with differences in measurement time yield discrepancies between measurements on the two platforms. I think this hypothesis could use some further analysis, given that winds are much stronger later in this 24 hr period, where there appear to be non-trivial differences between measurement times between the two platforms as well. Is there some kind of forcing at the mesoscale or synoptic scale (or something else?) to explain this condition, where the inter-platform differences early on are much more pronounced than later?  To me it seems plausible that this difference could be partly related to the platform heating issue that was shown in Figure 2.

[The difference below 100 m could be a result of the heating issue, but above that height, the sensors are very well aspirated and should not be influenced by the temperature of the shell. Since the winds measured by the radiosonde in that layer are >15 m/s coming from the south and there was a 30-minute difference in sampling time, the radiosonde was likely taking measurements dozens of kilometers to the north, where it was colder at the surface.

I would also disagree that we do not see those effects when the winds are stronger. Where the winds are strongest, we cannot compare the measurements due to the wind constraints on the CopterSonde. The authors input new figures to capture the thermodynamics and kinematics of each case. In the new Figure 3c and 3d, we can see the specific humidity is different beneath the jet. The moisture surge around 0930 UTC is weaker using the radiosonde data.]

L 230: Could the authors please clarify what is meant in this sentence beginning with 'Contrary'? Shapiro et al. (2016) indicate in their paper (pg 3045) that buoyancy is specified to be at maximum a few hours before sunset, also depicted in Figure 7. From references to this elsewhere in the paper, surface buoyancy peaks at three hours prior to sunset. Given a local sunset time of about 0010 UTC, my visual estimate is roughly a four-hour difference between the peak of vertically integrated buoyancy and the time of sunset in the authors' Figure 6. This seems reasonably close to the result in Shapiro et al. (2016), in contrast to the authors' statement on L 230. In addition, the authors are comparing maximum surface buoyancy of Shapiro et al. (2016) to their vertically integrated buoyancy expression. Could the authors please elaborate on why and under what circumstances these terms are directly comparable?

[The authors agree, that this statement is more in agreement than in contrast. Upon further inspection, it appears I had made a mistake in thinking I was referencing an older paper that used the sunset time as buoyancy max. Since the Shapiro et al. 2016 paper is more recent, I have changed the format of the sentence to express that our results agree with the model results (L 241-243). The authors also added a point of difference of the vertical buoyancy evolution from each study. It was not shown in the manuscript but the vertically integrated buoyancy evolution aligns very well with the surface buoyancy since the vertical buoyancy profiles are nearly constant in a CBL. This also seems to agree with Figure 7 from Shapiro et al. 2016.]

L 265: I'm unclear what the authors mean by "process continues throughout the night . . .". The timeline of Figures 9 and 10 begins around 0530 LT (MDT) and continues until around 1100 LT. In the following sentence, are the authors referring to the inversion as shown on Figure 9, which increases after sunrise and then remains fairly constant in time?

[This line is referring to the overall process of drainage flow, not either of the figures. L 282-285 were restructured to make the ordering of the process more explanatory.]

Figure 3: Qualitatively, the two subplots appear to match-up well except in the first few hours, roughly before 1900 UTC. The CopterSonde wind measurements appear to noticeably underestimate those of the radiosonde at these times between about 400 m and 900 m AGL, but also in the region of roughly 100 – 300 m around 2000 UTC. Corresponding buoyancy values in this spatio-temporal region are also show non-trivial discrepancy in the initial hours at the upper levels. The authors provide some brief analysis later in the text (L 211-213). The authors contend that the discrepancy is due to a difference in location between the two platforms. I noticed that coherence between the two platforms is fairly high later on in the period, when wind speeds are considerably stronger. Is there a known physical reason to expect why the mixed layer temperature should be so different in an adjacent space 30 minutes later given the relatively flat and homogeneous nature of the domain early on? Can these buoyancy differences be explained by the platform solar

heating problem (L 180-181)? Also, I would urge the authors to shift their analysis of Figure 3 earlier in the text, to where the Figure is first introduced.

[In regards to the observations before 1900 UTC, I think a large source of this discrepancy is that the radiosonde data is being interpolated across 3 hours of data that do not exist. The large difference in buoyancy aloft is a combination of the difference in cooler surface temperatures influencing the radiosondes 1432 UTC values and the compensation of the interpolation to bring the subsequent profile back to neutral between times of profiles. Looking around 1724 with each platform, the buoyancy profiles are pretty similar, as is the 2000 UTC one. So, those observations between 1432-1724 UTC assume a completely linear transition. This is especially untrue during transition periods, heterogeneity is very high. After 1724, the new Figure 3 shows strong agreement in potential temperature and specific humidity.

As for the wind speeds around 2000 UTC, I agree that there appears to be a non-trivial difference in the wind speed profile. Although, Bell et al. 2019 (https://doi.org/10.5194/amt-2019-453) showed that throughout the campaign the mean difference in wind speed from CopterSonde and the radiosonde data (and doppler lidar) is less than 1 m/s. The apparent difference is likely due to the binning on the colorbar is overexaggerating the difference in wind speeds. For example, the difference could be very small, say 11.9 m/s and 12.1 m/s, but binned in different colors to make it appear the difference is much larger. The authors kept the contour levels a bit larger to help with readability since the intercomparison of measurements is not the paper's goal.]

Technical Corrections:

L 20: decades old -> decades-old

[Fixed]

L 27: this type of data has –OR- these types of data have

[Fixed]

L 82: "Flux-Capacitor" may not be immediately recognized by the reader as the name of a campaign at this point in the manuscript. Suggest: "The Flux-Capacitor campaign" or "Flux-Capacitor (2018)", etc.

[Fixed]

L 83: Suggest: "Southern Plains" or "southern Plains" as the authors see fit – somewhere in here there's a proper noun and the authors are in a better position than the Reviewer to decide what that should be

[Fixed]

L 99: Suggest: San Luis Valley -> San Luis Valley, Colorado, U.S.A. given that AMT is an EGU journal.

[Fixed]

L 99, 111: Note that the same person (Gijs de Boer) is referenced as "de Boer et al." on L 99 and "Boer et al." on L 111. Suggest using the former naming format (and change to 2020a) and renaming the second reference as "de Boer et al. 2020b."

[Fixed]

L 103: Please define CASS here (first defined later on L 121)

[Fixed]

L 116: "the CopterSonde" hasn't yet been introduced at this point in the manuscript. Suggest replacing with " . . . utilized a rotary wing quadcopter RPAS . . ." or perhaps " . . .utilized a CopterSonde RPAS" for clarity.

[Fixed to "CopterSonde RPAS"]

L 124-125: ". . . pressure sensor within the autopilot." Do the authors mean that the pressure sensor is affixed to the autopilot board? Perhaps the authors could clarify here as the word autopilot is often used to refer to the software of the CopterSonde, too.

[The pressure sensor is built into the autopilot board to aid with altitude control. This explanation was added to that line.]

L 170: cross evaluated -> cross-evaluated

[Fixed]

L 176: long established -> long-established

[Fixed]

L 205-206: Doesn't this sentence effectively repeat the statement on L 201-203? Likely can omit.

[Fixed]

L 207-208: "the height of mixed potential temperature" -> "the height of the mixed layer"

[Fixed]

L 210: replace comma with semi-colon;

[Fixed]

L 226: Could the authors please include the equation used to compute vertically integrated buoyancy to verify the units as shown in Figure 6?

[It is not technically integrated, rather summed over the entire profile, which is why the original units are preserved. The authors have changed the phrasing from vertically integrated to vertically summed.]

L 234 -245: The verb tense changes over the course of these two paragraphs, from past to present to future. Please be consistent with verb tense throughout.

[Fixed]

L 260-261: Assuming that the authors mean that the buoyancy foretells of change to local convection strength (?), I would suggest 'portend', 'herald' or 'precede' as a replacement for 'predate.'

[Fixed]

L 269: Add 'UTC' after '1530'

[Fixed]

L 281: "below the ABL" -> below the top of the ABL (or similar)

[Fixed]

L 323: Note that AMT asks that authors include the DOI in the references where available (https://www.atmospheric-measurement-techniques.net/submission.html#references )

Figure 1: The caption indicates that the red lines are from the CopterSonde and the blue lines are from the parcel calculation, but the figure shows the opposite color assignment.

[Fixed]

Figure 1: Could the authors please enlarge the text of the legend in subplots (a) and (c) slightly to improve legibility.

[Nearly every plot was altered to increase readability and increase font size.]

Figure 1: Would suggest indicating the UTC times of the examples in the caption to help the reader distinguish the top and bottom rows of the figure.

[The figure titles were simplified and the time is in a more readable format]

Figure 2: In the caption, add a space between 2018 and Purple.

[Fixed]

Figure 2: The time of sunset (sunrise) in Washington OK on 05 October 2018 was approximately 1910 CDT (0727 CDT), or 0010 UTC (1227 UTC) (e.g.,: https://sunrise-sunset.org/us/washington-ok/2018/10 ). The sunset (sunrise) graph on the figure indicates approximately 2020 UTC (0121 UTC), however. Could the authors please revise the figure? Also, the abscissa tick mark labels do not appear to cover the time range indicated by the caption (1500 UTC 05 October 2018 to 1430 UTC 06 October 2018).

[Fixed]

Figure 2: The CopterSonde temperature graph (black) is drawn as a continuous line over this period (indicated to be approximately 24 hours according to the caption), but Table 1 indicates that these observations were taken during 46 flights averaging 30 minutes each. Why the black-shaded graph in this figure drawn this way? Could the authors please revise the figure to indicate where CopterSonde observations are present versus absent?

[The line was replaced with black points to reflect the non-contiguous measurements.]

Figure 3: In the caption, some text is missing before the word filled: e.g. …"and wind speed (m s$_{-1}$) is shown in filled contours"

[Fixed]

Figure 4: Could the authors please enlarge the text in the legend in the upper right corners of both subplots to improve legibility? Same for Figure 5 and other figures where this point size is used.

[Fixed]

Figure 4: Please indicate in the caption the date and location of these measurements as done in Figure 3.

[Fixed]

Figure 4: The graphs indicating time of sunset appears a bit off with respect to the abscissa tick mark labels in subplot (b). The time of sunset should be around 0010 UTC by my findings for 05 October 2018 in Washington, OK.

[Fixed]

Figure 5: In the caption: buoyancy determined -> buoyancy-determined

[Fixed]

Figure 7: Could the authors please add a ruler showing distance to these subplots? The authors make frequent reference in the adjacent text to distances from the stations and it would be helpful to have that reference on the subplots. Also, it is somewhat difficult to discern where the mountains are located with the radar reflectivity present and the presence of multiple colors shading the surface. Perhaps the authors could add a subplot showing relief? This is mostly in response to the statements on L 243-244 about issues with radar coverage and interference from mountains. I'm not entirely sure I follow this statement – are the authors referring to convective cell base height? I would anticipate convective cells within the valley would be recognized by radar if there were no obstructions between the cell and radar, no?

[The ruler was not included because it crowded the figure too much, as would a relief subplot. I have made reference to a paper which shows the aerial of the valley better in the methods (L 93-94). L 243-244 were rephrased stating that there is a lag in when the storms become visible to the radar because they must grow higher than the surrounding mountains. Since in some areas the mountains are 2 km higher than the valley floor, the towering cumulus stage immediately following convective initiation is obstructed by the mountains. It is not until the cell has entered its mature stage that it will grow taller than the surrounding mountains. Since this study was focusing on convective initiation, we relied in part on the ASOS station and personal accounts rather than the KPUX radar.]

Figure 7: Please add the label 'UTC' to the times labelled on the subplots, for clarity.

["UTC" was added to the subplots as suggested]

Figure 8: Suggest some rewording for clarity: "Unfilled contours are buoyancy (m s^-2) where solid (dashed) are positive (negative) and filled contours are specific humidity (g kg^-1) from CopterSondes . . . "

[Since the figure was changed, the caption was altered and I believe it is a bit more clear]

Figure 8: Date format in the subplot titles differs from the format used elsewhere in the manuscript (DD MMM YYYY).

[Fixed]

Figure 9 & 10: I would suggest merging these two figures as subplots of a single figure, as the authors did in Figure 8. The authors use both Figures 9 and 10 to describe the sequence of physical processes in the final paragraph of Section 5. I think it would be useful to have the figures side-by-side to facilitate comparison.

[The figures were combined and the references throughout changed as well.]

---

## Author Response (AR2)

Author Responses are in red

Referee 1 Report

The authors have considerably improved the manuscript and have addressed most of my suggestions and comments in an appropriate way. There are, however, a few minor issues left that I would like to see changed before the final publication.

The main point is that the figure labels of figures 1 to 6 are still too small and thus hard to read!
I tried to find a good balance of large axis and contour labels and frequency. This was the best balance I could manage. Most of the figures are eps files which maintain their high resolution even when zoomed in very far. Figures 3 and 8 were adjusted to make the contour labels more consistent.

RC: Line 16: sounds like only warm air is transported vertically, I suggest removing "warm" here
AR: "warm" was removed
no! It's still in the manuscript!
Warm air was replaced with momentum. My mistake, I must have edited the paragraph again after responding to that comment.

RC: Figure 1, profiles for 15:15 UTC (lower panels): I assume the lowest 200 m indicate the mixed layer of the developing CBL after sunrise. I am a bit puzzled over the super-adiabatic slope of the environmental curve over the whole BL/ML depth. I can't explain this other than by a systematic bias from your measurements, e.g due to sensor time lag!
AR: The superadiabatic layer only extends about 50 m. The remainder of the layer is dry adiabatic. Also, a systematic sensor bias would have implications throughout the entire profile causing a linearly propagating error. Radiosonde data also supports a shallow superadiabatic layer during the day (attach fig).
I have to apologize, I read the graphs as the potential temperature! My bad!
But it also brings up to my mind if plotting the potential temperature would be much easier, as static stability can immediately seen from the slope and not from a comparison with the dry adiabate?
Figure 1 uses temperature because that is the primary variable used to calculate buoyancy. The goal of Figure 1 is to display how buoyancy relates to the temperature profiles. In following figures potential temperature is used to relate to static stability more easily.

Referee 2 Report

Many thanks to the authors for thoroughly and thoughtfully addressing each of my major and minor comments with appropriate changes in the text.

Comments:

1) Just to raise awareness- Regarding the authors' reply to my comment about L 230 (version 1), it looks like the changes mentioned by the authors are approximately around L 266-267 (version 2), rather than L 241-243 as mentioned in the authors' reply. I agree with the location of the changes in version 2. There were some other mismatches between referenced changes and actual changes but I think was able to locate all of them.
My apologies, I will keep better track of lines in the future.

2) Figure 1: Could you add the time units (UTC) either to the title of each subplot or to the caption.
Done